# Antitumor Effects of an Anthocyanin-Rich Grain Diet in a Mouse Model of Lewis Lung Carcinoma

**DOI:** 10.3390/ijms25115727

**Published:** 2024-05-24

**Authors:** Maria A. Tikhonova, Olesya Y. Shoeva, Michael V. Tenditnik, Anna A. Akopyan, Ekaterina A. Litvinova, Nelly A. Popova, Tamara G. Amstislavskaya, Elena K. Khlestkina

**Affiliations:** 1Institute of Cytology and Genetics, Siberian Branch of the Russian Academy of Sciences (IC&G SB RAS), 630090 Novosibirsk, Russia; olesya_ter@bionet.nsc.ru (O.Y.S.); amstislavskayatg@neuronm.ru (T.G.A.); 2Scientific Research Institute of Neurosciences and Medicine (SRINM), 630117 Novosibirsk, Russia; 3Department of Neuroscience, V. Zelman Institute for Medicine and Psychology, Faculty of Life Sciences, Novosibirsk State University, 630090 Novosibirsk, Russia; 4N.I. Vavilov All-Russian Research Institute of Plant Genetic Resources, 190000 St. Petersburg, Russia

**Keywords:** bioflavonoids, functional food, wheat grain, cancer, metastases, mice, cytokines, blood serum, tumor-associated macrophages, autophagy

## Abstract

Functional foods enriched with plant polyphenol anthocyanins attract particular attention due to their health-promoting properties, including antitumor activity. We evaluated the effects of a grain diet rich in anthocyanins in a mouse model of Lewis lung carcinoma. Mice of the C57BL/6 strain were fed with wheat of near-isogenic lines differing in the anthocyanin content for four months prior to tumor transplantation. Although a significant decrease in the size of the tumor and the number of metastases in the lungs was revealed in the groups with both types of grain diet, the highest percentage of animals without metastases and with attenuated cell proliferation in the primary tumor were observed in the mice with the anthocyanin-rich diet. Both grain diets reduced the body weight gain and spleen weight index. The antitumor effects of the grain diets were associated with the activation of different mechanisms: immune response of the allergic type with augmented interleukin(IL)-9 and eotaxin serum levels in mice fed with control grain vs. inhibition of the IL-6/LIF system accompanied by a decrease in the tumor-associated M2 macrophage marker arginase 1 gene mRNA levels and enhanced autophagy in the tumor evaluated by the mRNA levels of Beclin 1 gene. Thus, anthocyanin-rich wheat is suggested as a promising source of functional nutrition with confirmed in vivo antitumor activity.

## 1. Introduction

Today, functional foods enriched with biologically active substances are considered an essential part of dietary therapy [1]. The issue of proper, balanced nutrition is becoming increasingly acute, and this topic is relevant for specialists in all areas of medicine. Of particular interest are products with special properties that can reduce the risk of developing certain diseases. Cardiovascular diseases and cancer have been leading the structure of population mortality for several decades. According to the International Agency for Research on Cancer (IARC) [2], the 5-year prevalence of cancer in the worldwide population is 53.5 million people, with almost 20 million new cases and 9.7 million deaths every year. Among this group of diseases, the leading cause in the structure of population mortality remains lung cancer, accounting for 18.7% of deaths. Despite the achievements made in recent decades in the field of oncosurgery, chemotherapy and radiation therapy, many tumors remain resistant to treatment. At the same time, the side effects of treatment of sensitive tumors lead to a lower quality of life. Hence, alternative methods aimed at preventing and reducing the risk of cancer and lung cancer in particular are in demand.

Information about the connection between diet and certain oncological processes has been demonstrated quite a long time ago and continues to accumulate [3]. It is believed that a third of deaths from malignant neoplasms could be avoided by modifying the structure of the diet by increasing the amount of vegetables and fruits [4]. Among the health-promoting chemicals, polyphenols attract particular attention as the most abundant reducing compounds ingested in the human diet [5]. These compounds are present in fruits, vegetables, spices, and beverages (red wine, tea, coffee), and their total intake was estimated to be approximately 1 g/day [6]. Among the consumed polyphenols, flavonoids account for two-thirds of the total intake, and phenolic acids account for the remaining one-third. The flavonoid compounds anthocyanins, which are easily recognized by their pigmentation varying from blue to deep purple [7], represent a regular component of the human diet and are considered one of the promising group of polyphenols for enriching functional foods. Both human and animal studies revealed that anthocyanins and concomitant polyphenols are functional compounds, which are able to diminish free radical damage, chronic inflammation and the risk of mutations, and they can attenuate the development and progression of chronic disorders, such as atherosclerosis, metabolic syndrome, eye and kidney complications, many cancer types, and they are also able to control body weight [8]. The health-promoting effects of the flavonoids are based on their antioxidant properties and ability to interact with cells’ proteins (receptors, kinases, or transcription factors) and modulate signaling pathways [9]. The main natural sources of these substances are dark-colored berries and fruit coverings. However, the use of grain products, in particular wheat, enriched with phytochemicals is one of the promising approaches to improving public health with the ubiquitous consumption of baked goods, which are accessible to all groups of the population and that are consumed regularly in sufficient amounts [10].

In recent years, breeding for the anthocyanins content in wheat grains has become a focus [11] as anthocyanin-rich cultivars have shown to be very promising as functional foods [12,13]. The development of biofortified colored wheat adds nutritional and functional health benefits to the energy-rich wheat. Colored wheat exists in three forms, purple, blue, and black, depending upon the types and positions of the anthocyanins in the wheat layers [14]. However, the information on the health-promoting effects of wheat anthocyanins is very scant [15]. Recently, higher radical scavenging and membrane protective activities of wheat bran extract with a high level of anthocyanins have been found using mouse tissue homogenates (brain and testes) and suspension of murine erythrocytes [16]. Using mouse models of Parkinson’s and Alzheimer’s diseases, the neuroprotective activity of wheat anthocyanins has been demonstrated [17]. Wheat grain enriched in anthocyanins significantly reduced the body weight gain and fat pad in mice fed with a high-fat diet [18]. Moreover, most of the studies on the antitumor activity of anthocyanins have been performed using their extracts in vitro and little is known about their in vivo effects.

This study aimed to evaluate the effects of a grain diet rich in anthocyanins in a mouse model of Lewis lung carcinoma (LLC) in C57BL/6 mice. We used two wheat lines that have similar genomes with the exception of a small part of chromosome 2A, which contains the *Pp3*-regulating anthocyanin biosynthesis [19]. The wheat near-isogenic lines differing in the grain anthocyanin content applied in the current study allowed for establishing the role of the *Pp3* gene that marks the line with anthocyanin-rich grains in protection against lung cancer and revealing some associated molecular mechanisms of in vivo anthocyanin action, including alterations in the serum cytokine profiles.

## 2. Results

As in our previous experiments [17], body weight was significantly influenced by a grain diet (Figure 1a). Having observed an increase in the body weight of mice over time during the experiment, we identified a significant effect of the type of diet (*F*(3, 34) = 37.7, *p* < 0.001), duration of feeding (repeated measurements) (*F*(4, 136) = 388.6, *p* < 0.001), as well as their interaction (*F*(12, 136) = 47.8, *p* < 0.001) on the body weight gain. The mice kept on grain diets had reduced body weight gain as compared to the mice fed with standard chow. If in the groups fed with standard chow a significant increase in the body weight occurred every month, then in the grain-fed groups an increase in the body weight occurred more slowly, once every two months, and it was significantly less than in the groups receiving standard food since the first month of feeding (Figure 1a). The experimental groups also differed significantly in the weight indices of the spleen normalized to the body weight (*F*(3, 31) = 34.6, *p* < 0.001; Figure 1b). In mice in the “LLC_St. diet” group (LLC model as a standard diet), the index was markedly augmented compared to mice of the Intact group (*p* < 0.001), while in mice fed with both types of grain diet (“LLC_CGr” (LLC model as a control grain diet) and “LLC_Gr_HCA” (LLC model as a grain diet with high anthocyanin content), it was significantly reduced compared to the mice in the “LLC_St. diet” group (*p* < 0.001), down to the values of the Intact mice.

Significant differences were revealed in tumor growth and metastasis between the groups with different types of diet. The effect of a diet on the tumor weight (*F*(2, 64) = 18.0, *p* < 0.001; Figure 2a) and the number of lung metastases (*F*(2, 64) = 6.6, *p* < 0.01; Figure 2b) was shown. In the mice kept on grain diets, the tumor mass was significantly less than in the group given standard chow (*p* < 0.001). The number of lung metastases was also significantly reduced in both groups fed with grain compared to the group given standard chow (Figure 2b). Noteworthily, the percentage of animals with metastases was substantially decreased in the “LLC_Gr_HCA” group compared to the group given standard chow (*p* < 0.01) and the “LLC_CGr” group (*p* < 0.05) (Figure 2c).

Moreover, significant differences were revealed in the level of tumor cell proliferation (*F*(2, 10) = 5.5; *p* < 0.05) between the groups of mice receiving different types of diet. In the group of mice fed with the wheat enriched with anthocyanins, the expression of the proliferation marker Ki67 was significantly lower compared to the “LLC_St. diet” and “LLC_CGr” groups (Figure 3). This result agrees with the significant decrease in the percentage of animals with metastases in the “LLC_Gr_HCA”.

To assess the systemic immune response to tumor transplantation in mice with different diets, the levels of 32 cytokines and chemokines in the blood serum were analyzed. The results are summarized in Table 1. Tumor transplantation resulted in an increase in the levels of TNFα, LIF, IP-10, KC, IL-6, IL-7, IL-10, MCP-1, MIG, MIP-1α, VEGF, and G-CSF, as well as a decrease in the level of eotaxin, in “LLC_St. diet” group vs. Intact mice. The groups with both types of grain diets did not differ from the group fed with standard diet in terms of the serum levels of TNFα, LIF, KC, IL-5, IL-7, IL-10, MCP-1, MIG, MIP-1α, and VEGF. Nevertheless, in the group that received grains enriched with anthocyanins, there was a significant increase in the levels of IP-10 and eotaxin, as well as a decrease in the levels of IL-6 (Figure 4a) and G-CSF, compared to the “LLC_St. diet” group. The “LLC_CGr” group had augmented levels of IL-9 and eotaxin, and decreased levels of G-CSF, compared to the “LLC_St. diet” group. It should be noted that an undetectable level of LIF was observed in 11.1% of mice in the “LLC_St. diet” group compared to 87.5% of Intact animals (*p* < 0.01). Moreover, there were no mice with an undetectable level of LIF (0%) in the “LLC_CGr” group, while in the “LLC_Gr_HCA” group, this index was 66.7% and it differed significantly from the “LLC_St. diet” (*p* < 0.05) and “LLC_CGr” (*p* < 0.05) groups (Figure 4b).

We also studied a local immune response, which was assessed by the expression of a number of genes in the tumor tissue. The mRNA levels of both markers of activated macrophages were reduced in both groups with grain diets. The expression of the *Arg1* gene encoding a marker of M2 microglia was significantly influenced by the type of diet (*H*(2, N = 27) = 11.03, *p* < 0.01) as well as the expression of the *Nos2* gene encoding M1 microglial marker (*H*(2, N = 28) = 8.75, *p* < 0.05). The *Arg1* mRNA levels were significantly diminished in the “LLC_Gr_HCA” group, while the *Nos2* mRNA levels were markedly reduced in both the “LLC_Gr_HCA” and “LLC_CGr” groups (Figure 5a,b). At the same time, the expression of the *Becn1* gene encoding an autophagy-related marker Beclin 1 was augmented in the grain-treated groups (*H*(2, N = 29) = 12.02, *p* < 0.01). A significant increase in the *Becn1* mRNA levels was observed in the “LLC_Gr_HCA” group (Figure 5c).

## 3. Discussion

Anthocyanins are a class of water-soluble polyphenols, which show a range of favorable effects for human health, such as the prevention of cardiovascular diseases, metabolic control, and antitumor activity. Their potential antitumor effects are related to a wide range of biological properties, including antioxidant; anti-inflammation; anti-mutagenesis; induction of differentiation; inhibiting proliferation by modulating signal transduction pathways, inducing cell cycle arrest, and stimulating apoptosis or autophagy of cancer cells; anti-invasion; anti-metastasis; reversing the drug resistance of cancer cells and increasing their sensitivity to chemotherapy [20]. However, studies of anthocyanins have mainly been carried out using extracts isolated from various products [21,22], and most of the studies were performed in vitro [23,24,25], while studies of products or raw materials that are used in the food industry are insufficient. Here, we assessed the effects of grain monodiets consisting of wheat from two lines, differing only in a small region of chromosome 2A, which carries the *Pp3* gene regulating anthocyanin synthesis in the grain pericarp, on a grafted immunologically compatible tumor LLC. It should be noted that the near-isogenic wheat lines used in this study exhibit no differences in the proteins, total dietary fiber content, metals, and amino acids content [26,27]. Moreover, the total phenolic content did not differ significantly between the lines, although the colored (referred to here as anthocyanin-containing) line displayed a slight increase compared to the non-colored (referred to here as control) line [26,27]. The anthocyanin levels in both the whole-grain and bran extracts were higher in the colored line compared to the non-colored line [28]; the content of anthocyanins in the Gr_HCA was as high as 140 mM/g [26]. High-performance liquid chromatography with mass spectrometry of the wheat extracts revealed that the anthocyanin-enriched grain contains two major substances, cyanidin glucoside and cyanidin arabinoside, as well as several minor anthocyanins, while no anthocyanins were found in the bran extracts of the control grain [16].

The LLC is a syngeneic model of lung cancer with good reproducibility. The LLC model is convenient as it can be used to assess both the growth of solid tumors and the extent of their metastasis. The tumor metastasizes according to the hematogenous type; it is believed that the only place of localization of metastases is the lungs. Metastases occur on days 6–8 after intramuscular transplantation. The LLC is considered similar to human lung cancer in terms of the sensitivity to antitumor drugs and is regarded as a model of non-small-cell lung cancer [29]. Syngeneic mouse models of non-small-cell lung cancer make it possible to use an immunocompetent host to evaluate the inflammatory and immunological factors that can influence tumor growth and metastasis.

At the first stage of the experiment, the mass of the tumor tissue was compared between groups of animals inoculated with a suspension of tumor cells. A total of 15 days after administration, the weight of the tumors in the groups receiving both types of wheat was lower than that in the group with regular laboratory chow. Our results agree well with previous findings about the antitumor activity of grain products. For instance, clinical epidemiological studies revealed that whole grain consumption could lower the breast cancer risk; moreover, certain components of wheat grain demonstrated antitumor activity in vitro, including growth inhibition potential against human breast and colon cancer cells [30]. 

Along with this, the body weight of the animals changed in a similar manner: the weight was significantly higher in the “LLC_St. diet” group compared to the grain-treated groups. Earlier, we found that mice of the C57BL/6 strain have been bearing both types of grain diet well up to six months of feeding, but both types of grain monodiets caused a significant decrease in body weight gain due to the lower food intake and lean diet containing only grain [17]. Here, we revealed that the delay in body weight gain has occurred since the first month of feeding with grain. Noteworthily, the mice treated with the grain with a high content of anthocyanins did not differ in the body weight gain from the mice fed with the control grain. Similar effects were observed in humans: a randomized Danish crossover trial revealed that the whole grain-rich diet reduces body weight [31].

One may suggest that the effect of reducing the tumor size is associated with calorie restriction. Many types of cancer cells are not able to execute changes that would allow survival in a nutrient-deficient and toxic environment [32]. Moreover, accumulated evidence suggests that calorie restriction can produce an adaptive autophagic response, while prolonged autophagy overstimulation leads to type II autophagic cell death [33]. At the early stages of tumor development, autophagy inhibits tumor growth, while later during the cancer progression, it may promote the survival of tumor cells [34]. We found a significant increase in the mRNA levels of the *Becn1* gene encoding Beclin1, one of the key autophagy-associated proteins, in the “LLC_Gr_HCA” group and a tendency to increase in the “LLC_CGr” group (*p* = 0.066). Probably, the autophagic response was more pronounced in mice treated with the grain with a high content of anthocyanins because of the additional autophagy stimulation, as anthocyanins can induce the autophagy of cells in an autophagy-related protein 5-dependent manner [35,36]; thus, an anthocyanin-enriched grain diet may produce a prolonged autophagic response, promoting the loss of cancer cells.

When calculating the number of metastases, a difference in this parameter was found between the “LLC_St. diet” and grain-treated groups. In the mice receiving a regular diet, the number of metastatic nodes was higher than in the groups with grain diets. At the same time, more intriguing was the fact that in the “LLC_Gr_HCA” group, about half of the animals had no metastases at all, while the LLC is a highly reproducible model that has gained widespread use precisely because of the almost 100% probability of metastasis from primary nodes. This phenomenon urged us to deeply explore the molecular events and processes that could be influenced by the anthocyanin-rich grain diet and account for the metastasis inhibition.

Indeed, the data obtained from the immunohistochemical detection of the proliferation marker Ki67 in the tumor tissue confirmed that the level of cell proliferation in the primary solid nodes of the LLC was significantly lower in the “LLC_Gr_HCA” group compared to both the “LLC_St. diet” and “LLC_CGr”, despite the fact that the tumor sizes in the grain-treated groups were comparable. These data are in good agreement with the previously identified effects of anthocyanins on cell proliferation. Anthocyanins can selectively inhibit the proliferation of cancer cells but have little influence on the proliferation of normal cells [37]. In vitro, anthocyanins can inhibit the growth and proliferation of cancer cells by inhibiting different kinases and other signaling pathways, such as β-catenin, Wnt and Notch, and regulating the expression of anti-oncogenes and relevant proteins [20].

Chronic inflammation is often a harbinger of a tumor. The abnormal overexpression and secretion of inflammatory factors are critical to tumorigenesis and metastasis. Tumor-associated macrophages are established as important regulators of tumor progression by impacting on tumor immunity, angiogenesis, and metastasis [38]. Metastasis of the LLC also involves the powerful stimulation of tumor-associated macrophages, which produce IL-6 and TNFα by activating TLR-2 and TLR-6. Both TLR-2 and TNFα are required for metastasis. At later stages, cancer cells interact with components of the host’s innate immunity, causing myeloid-derived progenitors to form an inflammatory microenvironment favorable for metastatic growth [39]. Tumor-associated macrophages generally exhibit an alternatively activated (M2) phenotype and promote the metastatic behavior of the LLC [40]. Here, we revealed a significant decrease in the mRNA levels of the marker of M2 macrophages *Arg1* in the tumor tissue in mice fed with grains enriched with anthocyanins but not in the group fed with control grain. This agrees well with the reduced percentage of mice with metastases in the “LLC_Gr_HCA” group. Both grain-treated groups had attenuated levels of the marker of M1 macrophages *Nos2* in the tumor tissue. Noteworthily, we earlier found that the anthocyanin-enriched grain diet substantially attenuated the activation of microglia, resident immune cells of the macrophagic type in the brain, in the nigrostriatal system and hippocampus in a mouse model of Parkinson’s disease [17]. Thus, the results evidence the inhibiting activity of the grain anthocyanins toward immune activation of cells of the macrophagic type.

In addition, an increase in the concentration of KC (CXCL1), a chemokine most potently expressed by tumor-associated macrophages [41], and IL-5, which stimulates cell migration by activating the ERK1/2 pathway, is a characteristic feature of malignant growth and metastasis [42]. The tumor process is accompanied by an increase in the cytokines MCP-1 and IL-10, although their role in the literature is designated as ambivalent in this process. Recent findings indicate that MCP-1 promotes the progression of tumors [43]. The multiplex immunofluorescence analysis performed here shed light on some features of the immune response in the groups treated with different types of diet. Although the studies described above were performed on other mouse models and some on cell models, in the present study, we observed a similar immune response for the LLC. The tumor transplantation resulted in an increase in the levels of TNFα, LIF, IP-10, KC, IL-6, IL-7, IL-10, MCP-1, MIG, MIP-1α, VEGF, and G-CSF, as well as a decrease in the level of eotaxin. An increase in the macrophage chemoattractant MIP-1a is associated with inflammatory events occurring in the tumor; an important role of this chemokine as a marker of bone resorption was noticed. Probably, the tumor inoculated intramuscularly had grown into a bone or produced an inflammatory process around the bone. A whole grain-rich diet was found to reduce systemic low-grade inflammation in humans, including a decrease in the serum inflammatory markers IL-6 and C-reactive protein [31]. Here, both types of grain diets produced a significant decrease in the levels of G-CSF, a colony-stimulating factor, which stimulates the survival, proliferation, differentiation, and function of neutrophil precursors and mature neutrophils. It should be noted that both types of the grain diet produced a significant decrease in the spleen index down to the values of Intact mice, indicating an anti-inflammatory effect of the grain diets.

The IL-6 cytokine has been described as having an important role in the development of malignant tumors [44], including in the growth of lung cancer-like tissue [45], as well as in the metastasis of malignant lung tumors [46]. Our study is consistent with these data. A decrease in its concentration was revealed in the “LLC_Gr_HCA” group, which was also characterized by an increase in the percentage of animals with an undetectable level of the LIF. LIF and IL-6 are cytokines of the same class and act unidirectionally, activating the STAT3 pathway, which stimulates the process of proliferation and migration in epithelial cells and aggravates the course of the malignant process [47]. LIF epigenetically blocks the chemokine MIG (CXCL9), which is involved in the polarization of macrophages along the M1 pathway and ensures the recruitment of cytotoxic CD8+ T cells into the tumor, which have an antitumor effect; hence, a decrease in the level of LIF causes tumor regression [48]. This is in good agreement with the results of this work, where animals from the “LLC_Gr_HCA” group showed a decrease in the IL-6 levels and an increase in the percentage of animals with undetectable LIF levels, along with a decrease in the tumor size and metastasis. In addition, the group that received an anthocyanin-enriched grain diet had augmented IP-10 (CXCL10) levels. Its antitumor activity is associated with chemoattracting several types of immune cells, such as cytotoxic T lymphocytes, NK cells, NKT cells, and macrophages, which infiltrate a tumor and prevent its growth [49]; CXCL10 promotes M1 polarization in inflammatory macrophages [50]. Moreover, CXCL10 is considered to inhibit non-small-cell lung carcinoma tumor growth and metastasis by disturbing tumor angiogenesis [51].

On the other hand, augmented levels of IL-9 were observed in the “LLC_CGr” group. IL-9 is secreted by a wide range of cells, but the most involved in this process are the helper T cell 9 and mast cells. The gene encoding IL-9 is considered a candidate gene responsible for the development of bronchial asthma. Eotaxin, an eosinophilic chemoattractant, attracts eosinophils, thus participating in the development of an allergic reaction. Its levels were significantly increased in both grain-treated groups compared to the group with the standard laboratory diet. The increased expression of IL-9 and eotaxin in the mice fed with the control grain suggests that the immunological response occurs according to the allergic type in the “LLC_CGr” group. The involvement of granulocytes and eosinophils has a cytotoxic effect on cancer cells but does not stabilize their proliferation. This is consistent with our data on the proliferative activity of cancer cells in this group, which did not differ from the “LLC_St. diet” group.

Thus, the effects of the control and colored wheat are associated with the activation of various immune mechanisms that cause a decrease in the tumor process. Since the difference between the wheat lines is the anthocyanin composition of the pericarp, we may conclude that the immune response switches under the influence of these substances. This assumption is consistent with recent findings about an inhibitory effect of anthocyanin delphinidin-3-O-glucoside and its metabolite delphinidin on the negative regulators, immune checkpoints PD-1 and PD-L1, resulting in the activation of an immune response in the tumor microenvironment and the induction of cancer cell death [52].

## 4. Materials and Methods

### 4.1. Experimental Animals and Procedures Involving Animals

The experiments were performed at the vivarium of the Scientific Research Institute of Neurosciences and Medicine (SRINM; Novosibirsk, Russia) using male mice of the inbred C57BL/6 strain born and reared under SPF conditions that were purchased from the SPF-vivarium of the Institute of Cytology and Genetics SB RAS (Novosibirsk, Russia, http://spf.bionet.nsc.ru/ accessed on 1 February 2024). The animals were housed in groups of 5–6 per cage (40 × 25 × 15 cm) under standard conditions (light–dark cycle: 14 h light and 10 h dark (lights off at 15:00); temperature: 18–22 °C; relative humidity: 50–60%). All the experimental procedures were carried out in accordance with the guidelines of the NIH Guide for the Care and Use of Laboratory Animals and were approved by the Inter-Institute Commission on Bioethics of the Institute of Cytology and Genetics of the Siberian Branch of the Russian Academy of Sciences No. 21.11, dated 30 May 2014, and directive 2010/63/EU. Every effort was made to minimize the number of animals used and their suffering.

The mice were subdivided into four groups and prescribed one of the following diets since the age of 6 weeks for 4.5 months; the experiments were conducted using a Lewis lung carcinoma (LLC) as an experimental tumor model. The mice of the Intact group received a standard granulated chow for laboratory mice (Ssniff R/M-H V1534-300, Soest, Germany) and pure water (Rosinka, Novosibirsk, Russia) ad libitum. The mice in the other three groups were transplanted with an LLC after four months of a diet. The mice in the “LLC_St. diet” group received a standard granulated chow for laboratory mice and pure water ad libitum. The mice in the “LLC_CGr” and “LLC_Gr_HCA” groups were subjected to a monodiet consisting of wheat grain of near-isogenic lines (i:S29*Pp-A1Pp-D1pp3*^P^ (Control Grain, CG) or i:S29*Pp-A1Pp-D1Pp3*^P^ (Grain with a High Content of Anthocyanins, Gr_HCA), respectively) and pure water ad libitum. he grain and standard chow were in free access and not limited. The line i:S29*Pp-A1Pp-D1Pp3*^P^ (Gr_HCA), marked by a dominant allele of the *Pp3* gene, accumulates anthocyanins in a grain pericarp, whereas the line i:S29*Pp-A1Pp-D1pp3*^P^ (CGr), characterized by a recessive allele of the *Pp3* gene, does not contain anthocyanins in the grain; the lines were developed at the Institute of Cytology and Genetics SB RAS (Novosibirsk, Russia) [19]. The content of anthocyanins in the Gr_HCA was 140 mM/g. The remaining elemental composition and amino acid content in the whole wheat flour obtained from the wheat lines were similar [26]. In the mice fed with the anthocyanin-enriched grain, the food intake per mouse during the 4.5-month-long feeding period was 416.5 ± 22.2 g and did not differ significantly from that of the mice fed with the control grain (433.6 ± 3.4 g; *p* > 0.05). The body weight gain was estimated monthly.

The LLC mouse model is the most common model of lung cancer; LLC cells remain tumorigenic and capable of metastasis into the lung in C57BL/6 mice. The transplantable tumors were obtained from the cell bank of the Institute of Cytology and Genetics (Novosibirsk, Russia). The LLC is maintained in solid form after intramuscular injection in C57BL/6 mice. The tumor tissue was minced by scissors, passed through a sieve 0.914 mm in diameter and resuspended in 0.9% NaCl prior to grafting. The LLC was injected into the right thigh of the mice at 800,000 cells in 0.1 mL per mouse [53,54]. 

The animals were sacrificed 15 days after the transplantation of tumor cells. The spleen and body masses were measured to evaluate a spleen index. Blood serum was collected for subsequent cytokine profiling using multiplex immunoassay; tumor tissue was taken for subsequent immunohistochemical analysis and determination of gene expression. Evaluation of the antitumor effect of the diets was based on the weight of the solid tumors and the number of tumor metastases in the lungs. Tumor metastases in the lungs were counted microscopically after their fixation in 10% formalin [53]. 

### 4.2. Immunological Multiplex Assay

To measure the cytokine levels, the blood of a mouse was collected into EDTA-containing tubes, and then within 30 min, the biosamples were centrifuged for 10 min at 2000 rpm; serum was taken and stored at −70 °C until assay. The cytokine concentrations were measured using a multiplex analyzer of proteins and nucleic acids (MILLIPLEX Luminex 200, Merck KGaA, Darmstadt, Germany) with xPONENT 3.1. software and xMAP technology with a Multiplex Assays reagent (MILLIPLEX MAP Mouse Cytokine/Chemokine Magnetic Bead Panel, Merck KGaA, Darmstadt, Germany) according to the manufacturer’s recommendations. The cytokine concentration was presented as the pg of the cytokine per mL of serum.

### 4.3. Immunohistochemical (IHC) Analysis

IHC analysis for a marker of proliferating cells Ki67 was performed in 30 μm thick frozen slices of the tumor tissue as described in detail earlier [55]. Primary antibodies against Ki67 (1:200; ab16667, Abcam, Cambridge, UK) and secondary goat-anti-rabbit Alexa Flur 488 antibodies (1:600; ab150077, Abcam, Cambridge, UK) were used to evaluate the Ki67 expression. The fluorescence images were finally obtained by an Axioplan 2 (Carl Zeiss, Oberkochen, Germany) imaging microscope and then analyzed with Image-Pro Plus Software 6.0 (Media Cybernetics, Rockville, MD, USA). The fluorescence intensity associated with the expression of Ki67 was measured as the background-corrected optical density (OD) with the subtraction of the staining signals of the non-immunoreactive regions in the images converted to grayscale. Negative control samples with the omitted primary antibody emitted only a minimal autofluorescent signal. Seven to ten sections of a tumor biosample were analyzed, and the mean fluorescence intensity was calculated. The area of interest was 18,192 μm^2^. For each image acquisition, all the imaging parameters were kept the same.

### 4.4. Gene Expression Analysis

We measured *Nos2* expression as a marker of the pro-inflammatory M1 and *Arg1* expression as a marker of the M2 subpopulation of macrophages as well as the expression of the *Becn1* gene encoding autophagy-related protein Beclin1 in the tumor tissue. The mRNA levels of the target and reference genes were measured by qPCR according to a previously published protocol [56]. The gene expression was normalized to the *Tubb5* (*tubulin*, *beta 5 class I*) mRNA level as ΔCt = 2 ^ (Ct *Tubb5* mRNA − Ct gene of interest mRNA). All the primer sequences were designed by our team using the Primer-BLAST (https://www.ncbi.nlm.nih.gov/tools/primer-blast/ accessed on 1 May 2020) [57] and Unipro UGENE version 1.32 [58] programs and published previously [56,59]. The following primer sequences were used for the real-time PCR analysis: *Tubb5*_F: TGAAGCCACAGGTGGCAAGTAT,*Tubb5*_R: CCAGACTGACCGAAAACGAAGT;*Arg1*_F: AAGAGCTGGCTGGTGTGGTG,*Arg1*_R: ACACAGGTTGCCCATGCAGA;*Nos2*_F: ATCGACCCGTCCACAGTATGT,*Nos2*_R: CATGATGGACCCCAAGCAAGA; *Becn1*_F: GAACTCACAGCTCCATTACTTA,*Becn1*_R: ATCTTCGAGAGACACCATCC.

### 4.5. Statistical Analysis

The normality of the data distribution was determined by the Shapiro–Wilk *W* test. For the comparison of metric variables grouped by one categorical variable, we applied the Mann–Whitney test and Kruskal–Wallis *H* test, followed by multiple comparisons of the mean ranks for all the groups in the case of the non-normal distribution of the data, or a paired Student’s *t*-test and one-way analysis of variance (ANOVA) for the normally distributed data. For the categorical variables (share of mice with metastases or with an undetectable level of LIF), the differences between the experimental groups were analyzed by Fisher’s exact criterion (2 × 2 contingency table).

Data were presented as the median and interquartile range (Q1; Q3) or mean ± S.E.M. The level of significance was defined as *p* < 0.05 in all the experiments reported here. STATISTICA 10.0 software (StatSoft, Tulsa, OK, USA) was used to perform all the statistical analyses.

## 5. Conclusions

The antitumor effects of the grain diet enriched with anthocyanins were more pronounced than those of the control grain diet. A significant decrease in the size of the tumor and the number of metastases in the lungs was revealed in the groups fed both types of grain diet, while the highest percentage of animals without metastases and with attenuated cell proliferation in the primary tumor were observed in the mice fed an anthocyanin-rich diet. Both grain diets caused a decrease in body weight gain in the mice. Although the reduction in body weight induced by a caloric restriction/fasting-mimicking diet could affect tumor growth, this does not appear to be the main mechanism of antitumor action of wheat grain of the different lines. Nevertheless, additive effects could not be excluded. Both grain diets produced a reduction in the spleen weight index, along with a decrease in the mRNA levels of the M1 macrophage marker *Nos2* in the tumor and a decrease in the G-CSF serum levels, indicating an anti-inflammatory effect of the grain diets. The antitumor effects of the grain diets were associated with the activation of different mechanisms. Mice fed with the control grain were characterized by an immune response of the allergic type with augmented IL-9 and eotaxin serum levels. The mice fed an anthocyanin-rich grain diet were characterized by reduced IL-6 serum levels, the highest percentage of animals with an undetectable LIF serum level, and augmented IP-10 serum levels. The related mechanisms may involve an inhibitory effect on the IL-6/LIF system, promoting the release of ligands of the CXCR3 receptor, chemokines CXCL9 and CXCL10 (IP-9), which chemoattract several types of immune cells that infiltrate a tumor and prevent its growth. Moreover, the anthocyanin-rich grain diet produced a decrease in the mRNA levels of *Arg1* encoding a marker of M2 tumor-associated macrophages, which are known to promote the metastatic behavior of the LLC, and enhanced autophagy in the tumor evaluated by the mRNA levels of *Becn1* encoding autophagic protein Beclin 1. Thus, anthocyanin-rich wheat is suggested as a promising source of functional nutrition with confirmed in vivo antitumor activity. Modulation of the systemic immune response and local immune processes, including cellular immune activity, contributes to the antitumor effects of wheat anthocyanins in vivo.

## Figures and Tables

**Figure 1 ijms-25-05727-f001:**
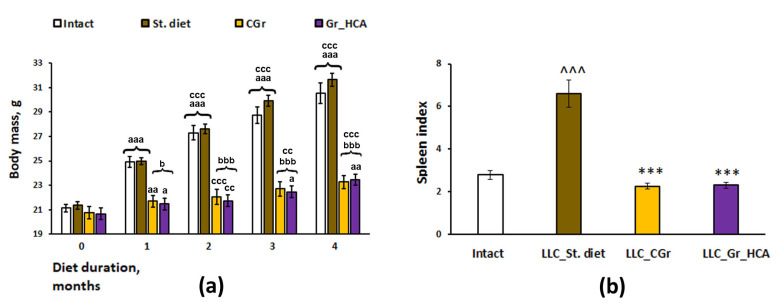
Effects of the type of diet on the body weight gain (**a**) and spleen index (**b**) in mice. Data are presented as the mean ± S.E.M. of the values obtained in an independent group of animals (n = 8–10 per group). Statistically significant differences: ^a^
*p* < 0.05, ^aa^
*p* < 0.01, ^aaa^
*p* < 0.001 vs. values of a respective group in a previous month; ^cc^
*p* < 0.01, ^ccc^ *p* < 0.001 vs. values of a respective group in the month before a previous month; ^b^
*p* < 0.05, ^bbb^
*p* < 0.001 vs. groups given a standard diet; ^^^ *p* < 0.001 vs. Intact group; *** *p* < 0.001 vs. “LLC_St. diet” group.

**Figure 2 ijms-25-05727-f002:**
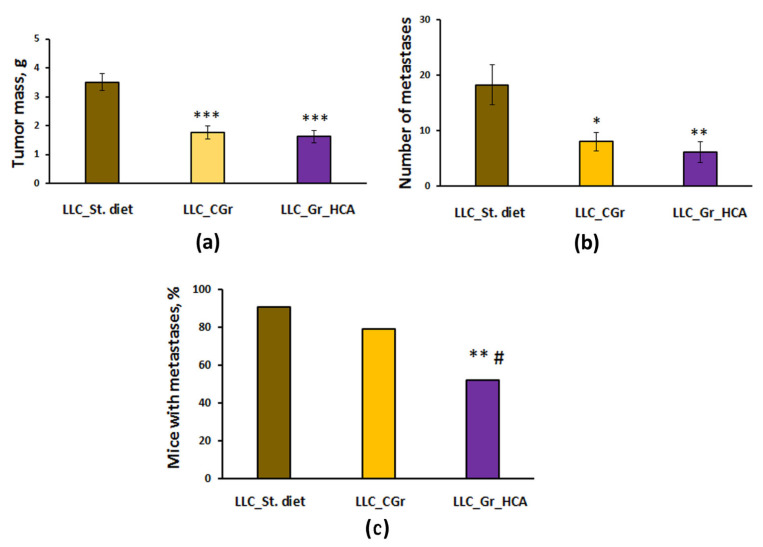
Effects of the type of diet on the tumor weight (**a**), number of metastases in the lungs (**b**), and percentage of mice with metastases (**c**) in an LLC model. Data are presented as the mean ± S.E.M. (**a**,**b**) or share of animals with metastases (**c**) of the values obtained in an independent group of animals (n = 21–24 per group). Statistically significant differences: * *p* < 0.05, ** *p* < 0.01, *** *p* < 0.001 vs. “LLC_St. diet” group; # *p* < 0.05 vs. “LLC_CGr” group.

**Figure 3 ijms-25-05727-f003:**
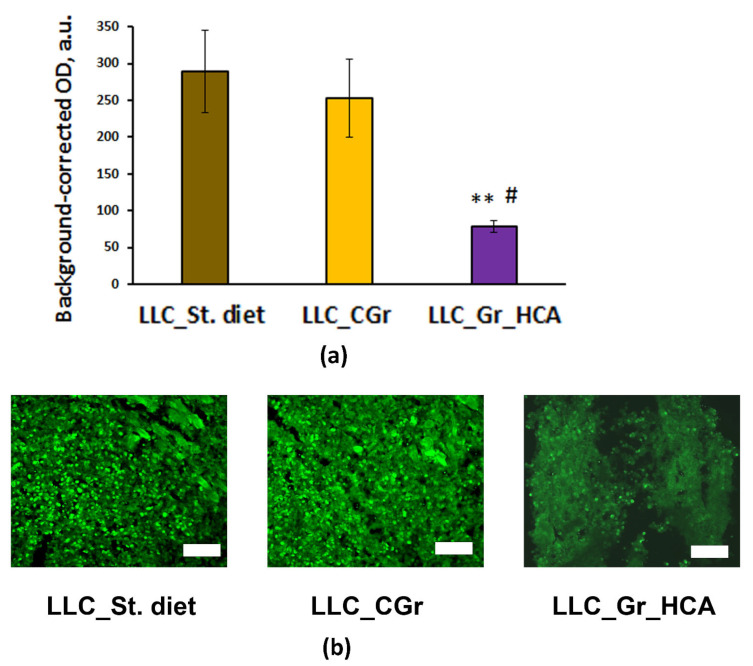
Effects of the type of diet on the expression of Ki67 in the tumor tissue of mice in an LLC model. (**a**) Quantitative results. Data are presented as the mean ± S.E.M. of the values obtained in an independent group of animals (n = 4–5 per group). Statistically significant differences: ** *p* < 0.01 vs. “LLC_St. diet” group; # *p* < 0.05 vs. “LLC_CGr” group. (**b**) Ki67 immunofluorescence in the tumor tissue. Alexa Flur 488 secondary antibodies were used for staining the samples. The fluorescence images were finally obtained by an Axioplan 2 microscope. Magnification: ×200; scale bar: 100 μm.

**Figure 4 ijms-25-05727-f004:**
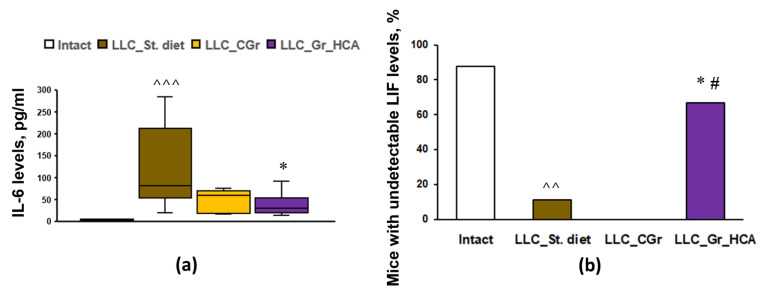
Effects of the type of diet on the serum levels of IL-6 (**a**) and the percentage of mice with undetectable serum levels of LIF (**b**) in an LLC model. Data are presented as the median (Q1; Q3) (**a**) or share of animals with metastases (**b**) of the values obtained in an independent group of animals (n = 6–9 per group). Statistically significant differences: ^^ *p* < 0.01, ^^^ *p* < 0.001 vs. Intact group; * *p* < 0.05 vs. “LLC_St. diet” group; # *p* < 0.05 vs. “LLC_CGr” group.

**Figure 5 ijms-25-05727-f005:**
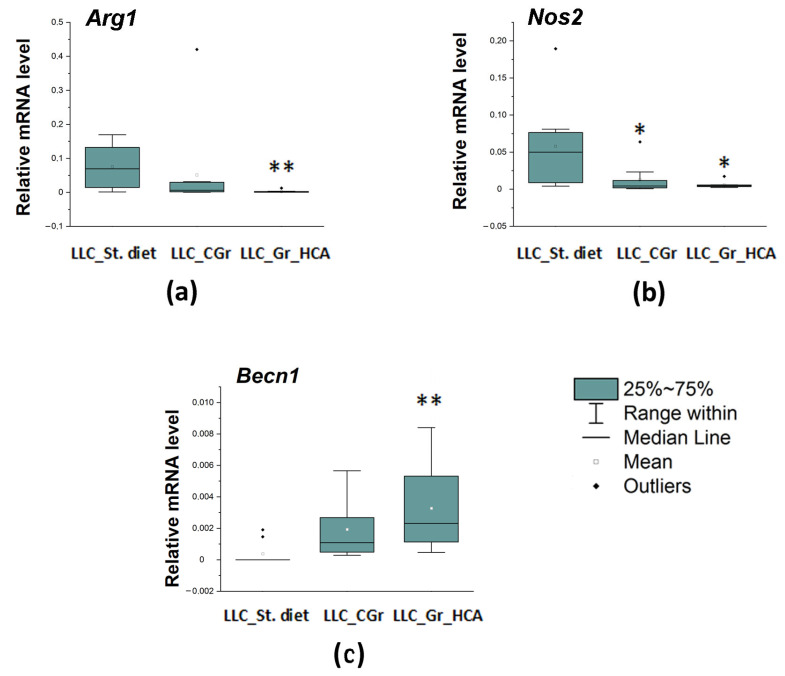
Effects of the type of diet on the mRNA levels of *Arg1* (**a**), *Nos2* (**b**), and *Becn1* (**c**) in the tumor tissue of mice in an LLC model. Data are presented as the median (Q1; Q3) of the values obtained in an independent group of animals (n = 9–10 per group). Statistically significant differences: * *p* < 0.05, ** *p* < 0.01 vs. “LLC_St. diet” group.

**Table 1 ijms-25-05727-t001:** Effects of the type of diet on the serum levels of cytokines and chemokines in the mice of an LLC model.

Analyte	Group				*F*, *p*/ *H*, *p* (LLC_St. Diet, LLC_CGr, LLC_Gr_HCA)
	Intact	LLC_St. Diet	LLC_CGr	LLC_Gr_HCA	
TNFα	4.6 (2.3; 5.4)	16.6 (13.5; 23.7) ^^	13.5 (12.4; 18.6)	15.3 (11.2; 37.9)	*H*(2, N = 20) < 1
IFNγ	0.89 (0.0; 1.29)	2.47(0.89; 5.44)	10.4 (3.2; 13.9)	0.86 (0.0; 2.47)	*H*(2, N = 20) = 5.23, *p* > 0.05
LIF	0.0 (0.0; 0.0)	1.6 (1.04; 3.11) ^^^	0.49 (0.13; 0.49)	0.0 (0.0; 1.6)	*H*(2, N = 20) = 3.37, *p* > 0.05
IP-10	121.5 (105.7; 150.5)	629.4 (435.5; 656.4) ^^^	847.7 (657.2; 1007.4)	858.4 (676.6; 1031.7) *	*H*(2, N = 20) = 7.14, *p* < 0.05
KC	75.9 (55.3; 109.2)	422.7 (190.9; 1119.5) ^	348.2 (95.7; 374.1)	238.6 (132.5; 554.1)	*H*(2, N = 20) < 1
IL-1α	111.7 (59.1; 146.5)	109.0 (81.6; 106.8)	114.6 (0.0; 178.4)	106.9 (78.3; 118.7)	*F*(2, 17) < 1
IL-1β	0.0 (0.0; 0.0)	0.0 (0.0; 1.28)	1.28 (0.0; 1.28)	0.0 (0.0; 0.43)	*H*(2, N = 20) < 1
IL-2	2.0 (0.0; 6.08)	3.18 (0.81; 3.18)	0.0 (0.0; 0.0)	3.18 (3.18; 3.18)	*H*(2, N = 20) = 2.43, *p* > 0.05
IL-3	0.07 (0.0; 0.34)	0.0 (0.0; 0.0)	0.0 (0.0; 0.0)	0.0 (0.0; 0.0)	*H*(2, N = 20) = 1.08, *p* > 0.05
IL-4	0.0 (0.0; 0.17)	0.0 (0.0; 0.0)	0.0 (0.0; 1.62)	0.0 (0.0; 0.0)	*H*(2, N = 20) = 6.32, *p* < 0.05
IL-5	2.96 (0.21; 6.69)	8.4 (6.8; 13.0)	11.6 (8.4; 30.9)	8.0 (0.4; 15.9)	*F*(2, 17) = 2.54, *p* > 0.05
IL-6	0.75 (0.28; 3.0)	80.3 (53.3; 194.6) ^^^	59.3 (20.5; 62.5)	29.9(21.9; 39.9) *	*F*(2, 17) = 3.33, *p* = 0.06
IL-7	0.07 (0.0; 0.3)	2.12 (1.04; 3.83) ^	0.53 (0.53; 0.53)	3.26 (0.07; 5.61)	*H*(2, N = 20) = 1.26, *p* > 0.05
IL-9	114.2 (34.7; 255.4)	145.1 (59.0; 190.5)	523.2 (339.6; 686.5) *	225.1 (100.3; 285.4)	*H*(2, N = 18) = 6.63, *p* < 0.05
IL-10	0.0 (0.0; 4.07)	30.5 (11.3; 129.3) ^^^	34.5 (4.07; 58.4)	33.4 (8.2; 131.2)	*H*(2, N = 20) < 1
IL-12p40	1.31 (1.31; 7.62)	4.7 (1.31; 18.9)	1.31 (0.0; 1.31)	0.0 (0.0; 0.0)	*H*(2, N = 20) = 4.92, *p* = 0.086
IL-12p70	3.8 (0.0; 10.7)	13.9 (7.5; 30.6)	13.9 (0.0; 55.3)	3.8 (0.0; 7.5)	*H*(2, N = 20) < 1
IL-13	80.5 (56.9; 91.6)	91.6 (60.3; 100.5)	87.2 (73.8; 91.6)	87.2 (73.8; 91.6)	*H*(2, N = 20) < 1
IL-15	16.5 (3.0; 37.7)	37.7 (27.1; 48.4)	16.5 (5.9; 27.1)	3.0 (0.0; 27.1)	*H*(2, N = 20) = 2.87, *p* > 0.05
IL-17	1.21 (0.16; 4.09)	0.93 (0.0; 2.58)	0.0 (0.0; 2.05)	0.0 (0.0; 0.0)	*H*(2, N = 20) = 3.79, *p* > 0.05
LIX	1584.2 (780.9; 2531.5)	1330.6 (227.4; 1711.1)	3155.6 (561.5; 3710.4)	1347.4 (184.5; 2807.1)	*F*(2, 17) = 1.56, *p* > 0.05
MCP-1	17.7 (0.0; 23.4)	420.7 (172.0; 3448.9) ^^^	356.3 (252.9; 1367.6)	759.6 (189.1; 2959.2)	*H*(2, N = 20) < 1
MIG	35.8 (30.6; 45.6)	166.4 (115.7; 227.3) ^^^	327.2 (68.6; 369.6)	202.3 (147.1; 263.8)	*H*(2, N = 20) < 1
MIP-1α	18.7 (0.0; 37.5)	82.4 (82.4; 129.6) ^^	82.4 (82.4; 119.7)	73.4 (0.0; 138.6)	*H*(2, N = 20) < 1
MIP-1β	55.7 (35.6; 62.7)	78.7 (74.1; 111.1)	66.6 (52.8; 74.1)	81.4 (58.6; 121.6)	*H*(2, N = 20) = 1.72, *p* > 0.05
MIP-2	18.7 (9.3; 167.7)	18.7 (0.0; 146.8)	146.8 (104.8; 218.8)	0.0 (0.0; 146.8)	*H*(2, N = 20) = 3.78, *p* > 0.05
RANTES	21.7 (9.4; 25.4)	17.9 (9.6; 22.9)	17.6 (8.1; 17.6)	24.8 (15.7; 26.8)	*F*(2, 17) = 1.95, *p* > 0.05
Eotaxin	1403.1 (865.3; 2269.9)	839.2 (525.5; 1166.6) ^	1365.4 (1211.6; 1608.4) *	1252.9 (804.7; 1378.2) *	*F*(2, 17) = 3.68, *p* < 0.05
VEGF	0.58 (0.19; 0.91)	1.43 (0.87; 2.17) ^	0.87 (0.48; 1.8)	0.91 (0.68; 1.34)	*H*(2, N = 20) < 1
M-CSF	8.2 (4.6; 11.7)	8.2 (5.8; 1122.1)	8.2 (8.2; 10.5)	3.4 (0.96; 3.4)	*H*(2, N = 20) = 6.22, *p* < 0.05
G-CSF	217.4 (192.3; 260.3)	2711.5 (1898.8; 4267.6) ^^^	788.2 (626.5; 819.2) **	1210.2 (703.9; 2598.6) *	*F*(2, 17) = 5.28, *p* < 0.05
GM-CSF	27.1 (20.6; 29.9)	27.1 (20.6; 32.8)	32.8 (0.0; 37.9)	10.3 (0.0; 27.1)	*H*(2, N = 20) = 1.2, *p* > 0.05

Data are presented as the median (Q1; Q3) of the values obtained in an independent group of animals (n = 6–9 per group). Statistically significant differences: ^ *p* < 0.05, ^^ *p* < 0.01, ^^^ *p* < 0.001 vs. Intact group; * *p* < 0.05, ** *p* < 0.01 vs. “LLC_St. diet” group. Abbreviations: G-CSF: granulocyte colony-stimulating factor; GM-CSF: granulocyte-macrophage colony-stimulating factor; IFNγ: interferon gamma; IL: interleukin; IP-10: interferon gamma-induced protein 10; KC: keratinocyte chemoattractant; LIF: leukemia inhibitory factor; LIX: lipopolysaccharide-induced CXC chemokine; MCP-1: monocyte chemoattractant protein-1; M-CSF: macrophage colony-stimulating factor; MIG: monokine induced by gamma interferon; MIP: macrophage inflammatory protein; RANTES: regulated on activation, normal T cell expressed and secreted chemokine; TNFα: tumor necrosis factor alpha; VEGF: vascular endothelial growth factor.

## Data Availability

The data presented in this study are available on request from the corresponding author. The data are not publicly available due to technical and time limitations.

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
