# Peer review of "Antitumor Effects of an Anthocyanin-Rich Grain Diet in a Mouse Model of Lewis Lung Carcinoma"

_ijms, 2024, doi:10.3390/ijms25115727_

Round 1
Reviewer 1 Report
Comments and Suggestions for Authors
Author Response
We would like to thank the Reviewer for his/her valuable comments and suggestions. We greatly appreciate the high esteem of our study. We have thoroughly revised the manuscript considering all the comments, which helped us to improve the manuscript. All major corrections made in the text are highlighted with green color. We believe that the revised version would be more clear and interesting for the readership of the journal.
- There must be quantitative determination of the anthocyanin content to be able to standardize the diet content used in this study and to be able to build up on these data. Also, qualitative determination of the anthocyanins present either by HPLC analysis with authentic compounds or through LCMS identification is necessary for the same reason. It is not known which anthocyanin and how much to use to get the same results.
Such analysis was performed earlier. We added the information about the content of individual anthocyanins and sum of anthocyanins in the grain with references to the Methods (L. 426-428) and Discussion (L. 230-241).
- Have the authors investigated histopathology changes?
We studied the expression of proliferation marker Ki67 in the tumor tissue (Figure 3).
- Tables should be numbered and cited in the text.
We added a number to the Table and cited it in the text as Table 1.
- Table of (the Effects of the type of diet on the serum levels of cytokines and chemokines in mice of a LLC model): It is not clear whether it includes standard error or not. What does (0.0) means? What do the numbers in brackets represent? More explanation is required.
The data in the Table are presented as the Median and interquartile range (Q1; Q3). This is indicated in the Table legend.
- Materials and methods: The accession numbers of the primers used must be mentioned.
All primer sequences were designed by our team using Primer-BLAST (Ye et al., 2012) and Unipro UGENE (Okonechnikov et al., 2012) programs and published previously (Feofanova et al., 2022; Belichenko et al., 2023). We added this information to Materials and Methods.
Reviewer 2 Report
Comments and Suggestions for Authors
The article under review evaluated the effect of an anthocyanin-rich cereal diet in a mouse model of Lewis lung cancer. Mice were fed wheat of near isogenic lines differing in anthocyanin content for four months before tumour transplantation. While the topic is interesting, I have serious reservations about the methodology and scope of the study.
Specific comments:
- It is essential to perform chromatographic analyses on the content of individual anthocyanins in the samples, or at least to determine the sum of anthocyanins and the sum of polyphenols. Without examining these contents, we are unable to determine what effect enrichment of the diet with this group of polyphenols (the effect of the dose of these active compounds) has on the parameters studied.
- The authors fail to state how much food the animals received per day and how many times they were fed, which is crucial information.
- Furthermore, the authors characterise the groups into which the mice were divided (LLC_St. diet, LLC_St. diet, LLC_CGr and LLC_Gr_HCA) only in the methodology at the end of the article. It is essential to explain the abbreviations denoting the different groups at the beginning of the article to facilitate analysis of the results.
I believe that the article will be suitable for publication only after it has been supplemented with the results of the studies described above.
Author Response
We would like to thank the Reviewer for his/her valuable comments and suggestions. We have thoroughly revised the manuscript considering all the comments, which helped us to improve the manuscript. All major corrections made in the text are highlighted with green color. We believe that the revised version would be more clear and interesting for the readership of the journal.
- It is essential to perform chromatographic analyses on the content of individual anthocyanins in the samples, or at least to determine the sum of anthocyanins and the sum of polyphenols. Without examining these contents, we are unable to determine what effect enrichment of the diet with this group of polyphenols (the effect of the dose of these active compounds) has on the parameters studied.
Such analysis was performed earlier. We added the information about the content of individual anthocyanins and sum of anthocyanins in the grain with references to the Methods (L. 426-428) and Discussion (L. 230-241).
- The authors fail to state how much food the animals received per day and how many times they were fed, which is crucial information.
Grain and standard chow were in free access and not limited (ad libitum). We had measured the food intake of mice and added the information on average food intake per mouse to Methods (L. 428-431).
- Furthermore, the authors characterise the groups into which the mice were divided (LLC_St. diet, LLC_St. diet, LLC_CGr and LLC_Gr_HCA) only in the methodology at the end of the article. It is essential to explain the abbreviations denoting the different groups at the beginning of the article to facilitate analysis of the results.
Done.
Reviewer 3 Report
Comments and Suggestions for Authors
Dear Authors,
Submitted Manuscript describes research on the effects of an anthocyanin-rich diet on the development of Lewis lung cancer. The Authors observed that in a mouse model in which this type of diet prevailed before tumor transplantation, there was a higher percentage of metastasis-free animals and attenuated cell proliferation in the primary tumor. Moreover, the authors concluded that such a diet is a promising source of functional nutrition with proven in vivo effects.
The experiments described in this Manuscript are well planned, well thought out, and the Methods are sufficiently described. However, the Article requires minor revisions before publication. The corrections mainly concern the section with the described results.
Figures 1,2,3,4 are illegible due to the undersized font used on the graphs. The reader has trouble interpreting the results shown.
Figure 3 - the fluorescence microscopy images are too small, making them unreadable. They should also include the magnification scale of the image (it is possible that she is in the lower right corner, but it is difficult to assess this due to the quality of the images). The description should also include the dye used in the staining, as well as the name of the microscope and the zoom under which the image was evaluated.
Table - it should be given a number and placed in the text. Below it should be a legend including all possible expansions of the abbreviations used in it. In addition, this table and the results presented in it are illegible. I am not convinced by this way of presenting the results. Perhaps it would be possible to find a solution and present it graphically?
Best regards
Author Response
Dear Reviewer,
we would like to cordially thank you for the careful review of our manuscript and providing valuable comments and suggestions. We greatly appreciate your high esteem of our study. We have thoroughly revised the manuscript considering all the comments, which helped us to improve the manuscript. All major corrections made in the text are highlighted with green color. We believe that the revised version would be more clear and interesting for the readership of the journal.
- Figures 1,2,3,4 are illegible due to the undersized font used on the graphs. The reader has trouble interpreting the results shown.
We have revised all the Figures and used larger size of the font.
- Figure 3 - the fluorescence microscopy images are too small, making them unreadable. They should also include the magnification scale of the image (it is possible that she is in the lower right corner, but it is difficult to assess this due to the quality of the images). The description should also include the dye used in the staining, as well as the name of the microscope and the zoom under which the image was evaluated.
We have revised the Figure and its legend according to the comment. Scale bar is in the lower right corner of each microphotograph. Please, refer to the Figure in the Word version of the manuscript or original high-resolution Figure uploaded separately but not to .pdf version as the quality and resolution of an image are significantly reduced upon converting to .pdf format.
- Table - it should be given a number and placed in the text. Below it should be a legend including all possible expansions of the abbreviations used in it. In addition, this table and the results presented in it are illegible. I am not convinced by this way of presenting the results. Perhaps it would be possible to find a solution and present it graphically?
We added a number to the Table and list of abbreviations used in the Table 1 to its legend. Here, we assayed 32 cytokines in four experimental groups. We suggest that such a big number of parameters and results would be more comprehensive in a form of a table. Nevertheless, we added a Figure (Figure 4) of the most significant results on the IL-6 levels and the percentage of animals with undetectable levels of LIF to illustrate and emphasize them.
Reviewer 4 Report
Comments and Suggestions for Authors
Dear author:
I have found the article under review to be highly intriguing, particularly in its exploration of the mechanisms underlying the inhibition of anthocyanins in wheat, and its potential implications for various diseases. While the findings presented are indeed relevant and valuable, there are a few areas that warrant attention:
Issues:
1. Image Resolution: The image resolution throughout the article is notably low, resulting in a lack of clarity. For instance, the legend in Figure 4 is nearly illegible. It is strongly advised to re-upload the original images to ensure proper visibility and comprehension
2. Analysis and Discussion Language: The language employed in the sections dedicated to results analysis and discussion tends to be excessively verbose, detracting from the depth of the content. Therefore, a more thorough and concise analysis of the data is warranted to enhance the clarity and impact of the findings.
3. Sample Size and Experimental Design: The number of mice allocated to each experimental group appears to be limited to 20-24, indicating potential randomness and insufficient sample size. Moreover, it is advisable for each group to encompass at least three parallel experimental sets to mitigate errors. Hence, I recommend supplementing the experimental data to strengthen the robustness of the conclusions.
Addressing these issues will undoubtedly bolster the quality and rigor of the manuscript, thereby enhancing its contribution to the scientific literature.
Sincerely

Comments on the Quality of English LanguageSatisfy the requirements for writing scientific and technical papers and be able to express research correctly.
Author Response
Dear Reviewer,
we would like to thank you for the careful review of our manuscript and providing valuable comments and suggestions. We greatly appreciate your high esteem of our study. We have thoroughly revised the manuscript considering all the comments, which helped us to improve the manuscript. All major corrections made in the text are highlighted with green color. We believe that the revised version would be more clear and interesting for the readership of the journal.
- Image Resolution: The image resolution throughout the article is notably low, resulting in a lack of clarity. For instance, the legend in Figure 4 is nearly illegible. It is strongly advised to re-upload the original images to ensure proper visibility and comprehension.
We have revised all the Figures and used larger size of the font. Please, refer to the Figure in the Word version of the manuscript or original high-resolution Figure uploaded separately but not to .pdf version as the quality and resolution of an image are significantly reduced upon converting to .pdf format.
- Analysis and Discussion Language: The language employed in the sections dedicated to results analysis and discussion tends to be excessively verbose, detracting from the depth of the content. Therefore, a more thorough and concise analysis of the data is warranted to enhance the clarity and impact of the findings.
We cannot agree with this comment. We suppose that the detailed information about statistical analysis should be provided in the Results section while Discussion should provide the results interpretation which would be supported current knowledge of the problem studied. Brief conclusions are given to summarize and enhance impact of the findings.
- Sample Size and Experimental Design: The number of mice allocated to each experimental group appears to be limited to 20-24, indicating potential randomness and insufficient sample size. Moreover, it is advisable for each group to encompass at least three parallel experimental sets to mitigate errors. Hence, I recommend supplementing the experimental data to strengthen the robustness of the conclusions.
Indeed, the number of mice allocated to each experimental group appears to be limited to 20-24. Nonetheless, we used inbred C57BL/6 strain born and reared at SPF conditions and well reproduced LLC model that provides low intergroup variability. For the most parameters studied, the amount of 5-10 inbred mice per group is considered as sufficient. This study encompassed two experimental sets. According to bioethical recommendations, we try to minimize the number of animals used and their suffering. Thus, we suppose that supplementing the experimental data is not reasonable in this case.
Round 2
Reviewer 1 Report
Comments and Suggestions for Authors
The authors have addressed all the comments.
Reviewer 2 Report
Comments and Suggestions for Authors
Despite my earlier decision to reject the article, I accept all the amendments. The Authors have addressed all my concerns.